# Study on the Manipulation Strategy of Metallic Microstructures Based on Electrochemical-Assisted Method

**DOI:** 10.3390/mi13122151

**Published:** 2022-12-05

**Authors:** Dongjie Li, Mingrui Wang, Weibin Rong, Liu Yang, Donghao Xu, Yu Zhang

**Affiliations:** 1Key Laboratory of Advanced Manufacturing and Intelligent Technology Ministry of Education, School of Mechanical and Power Engineering, Harbin University of Science and Technology, Harbin 150080, China; 2Heilongjiang Key Laboratory of Complex Intelligent System and Integration, School of Automation, Harbin University of Science and Technology, Harbin 150080, China; 3State Key Laboratory of Robotics and System, School of Mechatronics Engineering, Harbin Institute of Technology, Harbin 150001, China

**Keywords:** micromanipulation, electrochemical, manipulation strategy, finite element simulation

## Abstract

Microcomponent manipulation (MCM) technology plays a decisive role in assembling complex systems at the micro- and nanoscale. However, the existing micromanipulation methods are difficult to widely apply in the manufacturing of microelectromechanical systems (MEMSs) due to the limited manipulation space and complex application objects, and the manipulation efficiency is relatively low, which makes it difficult to industrialize these micromanipulating systems. To solve the above problems, this paper proposes an efficient metal MCM strategy based on the electrochemical method. To verify the feasibility and repeatability of the strategy, the finite element model (FEM) incorporating the hydrodynamic and electrochemical theories is used to calculate the local stress distribution of the contact position during the dynamic pick-up process. Based on the simulation results, we defined the relationship between the parameters, such as the optimal manipulating position and angle for picking, transferring and releasing. The failure behaviors of pick-up are built to realize the efficient three-dimensional manipulation of microcopper wire of 300 μm. By establishing a theoretical model and experimental verification, it was concluded that the middle point was the best manipulating position when picking up the microcopper wire, the most efficient picking angle was between 45 and 60 degrees for the pipette, and the average time was 480 s in three sets of picking–release manipulation experiments. This paper provides an achievable idea for different types of micro-object manipulations and promotes the rapid application of micromanipulation techniques in MEMSs.

## 1. Introduction

For decades, miniature manipulation has been a fundamental subject and hot exploration field [1,2,3] that has been a broadly considered by worldwide researchers. In numerous areas, dealing with proficiency and accuracy of miniature manipulation is turning out to be progressively significant [4]. Micromanipulation uses external field energy or power, microscale design and gadgets to accomplish push and pull, pick and release, position and different activities for different manipulation tasks. Be that as it may, the conventional metal miniature part control chiefly utilizes miniature clasping gadgets and bonds to accomplish pick-up and release. One of the main problems with the traditional micromanipulation method is the high requirement of the manipulation object and the low efficiency of a long time to complete a single manipulation.

Chu Jinkui et al. [5] have fostered a miniature cinch driven by electric intensity, which controls for the most part the veins in the measurement of 80 μm and cyanobacteria microorganisms with a width of 9 μm; Beyeler et al. [6] have studied miniature cinching techniques for electrostatic-driven incorporated capacitive sensors. The assortment and situating tests were completed on glass microspheres with a width of 35 μm and Hela cells with a measurement of 25 μm. Kenji Inoue et al. [7] designed a micromanipulation system with 8 degrees of freedom, which enables position adjustment and pick-up of spherical soft objects with a diameter of about 120 μm by using each degree of freedom in the control system as a manipulating strategy; B. Tamadazte et al. [8] designed a five degrees of freedom microassembly robot to automatically assemble micron-scale silicon particles with a positioning error of 1.4 μm and attitude error of 0.5 degrees. The above studies have achieved good performance based on designing microgrippers to manipulate round objects such as cells. With the improvement of microgrippers, the requirements for the size of the operating object have become progressively more flexible. However, the simple application of this technique in the MEMS case is challenging, and the control efficiency is low due to the space as well as micro-object shape constraints [9,10]. In terms of adhesion, Fan Zenghua et al. [11] have studied a micromanipulation method using surface condensation in which the manipulating tool integrates the refrigeration device, can condense water vapor in the air around the end of the probe and can liquefy into a liquid at the end of the manipulating tool, and the capillary force generated by the liquid bridge can be used to pick and release polystyrene microspheres with a radius of 20–25 μm. The operation is ideal but requires more space for operation. Regarding micromanipulation release, Gualtiero Fantoni et al. [12] studied a microclamping device made of elastomer film for hydrophobic partition. Before manipulation, the droplets go through mechanical stretching or contraction to achieve a change in the size of the hydrophobic area, to change the size of the contact angle between the micro-object and the gripper, to change the size of the liquid bridge force and to achieve the micro-object pick-up and release manipulation. This study requires a strict device for manipulation and has strict requirements for the material of the manipulated object. Xia Zhenhai’s team studied the doped graphene interface, which collects ambient humidity to form a large number of water bridges between graphene and the target surface, leading to a sharp change in adhesion force in the purpose of releasing objects [13]. However, the humidity requirement of the environment is strict and cannot be used for general application scenarios. Wang Lefeng et al. [14] has developed a Piezoelectricity-based capillary microcomponent transfer tool through the piezoelectric ceramic drive to apply voltage pulses and release trace droplets in the capillary tube in order to achieve a reliable release of micro-objects. The efficiency is low in the pick-up, and the release distance of the object is required for release. Zhang Qin et al. [15] developed a multineedle operator, which can change the shape of the droplet by adjusting the relative position between the control probes, so as to realize the position adjustment and release of the microresistance with the size of 1.6 mm × 0.85 mm × 0.5 mm. The structure of the operator is complicated and difficult to realize through industrialization.

The above micro-operation methods are all designed for grippers to achieve micro-object manipulation of a single shape. There are more requirements for operation space, operation object shape and size, and it is difficult to really apply to industrialization. Considering the above results, our group proposed a micrometer-scale metal manipulation method based on electrochemical principles [16]. This manipulation method is not a single design for the end gripper compared with the traditional micromanipulation method. This uses the deposition and electrolysis principles in electrochemistry to manipulate micron-sized metals of different sizes with high operational efficiency and no pollution to the environment, which can be widely used in the production and processing of microelectromechanical systems. It is an important contribution to accelerate the development of micro- and nanoscale industries. The metal microcopper wire is used as the manipulating objects and is placed on a metal substrate made of conductive silicon. The electrolyte used in the experiments is a dilute copper sulfate solution, and the pipette is made of quartz glass. The micromotion control stage is connected to a quartz pipette containing an electrolyte, and the pipette is controlled by manipulating the micromotion stage. The microcopper wire is located on the surface of the silicon substrate. The micromanipulation platform has a microscope tube, and a digital camera is mounted on top of the microscope. During the experimental manipulation, the experimental process is transmitted to the monitor through the microscopic vision system to view the micromanipulation process in real-time. The overall experimental platform and partial amplification are shown in Figure 1.

In addition, the contact angle between the manipulation tube and the manipulated object, the manipulation position, and the shape of the nozzle opening has an essential influence on the success rate of manipulation. If the manipulation point is not selected correctly during pick-up, it is easy to produce the slippage situation shown in Figure 2, leading to manipulation failure and easy damage to the manipulation pipette.

However, holding the pipette perpendicular to the object being manipulated is not the best way to manipulate. As shown in Figure 3, when the pipette is tilted α or β, it can also pick up the manipulation object well and may make release more straightforward. Therefore, based on the above analysis, the manipulation angle and the effect of the manipulation position on the manipulation success rate of the microcomponents represented by the micro-Cu and Pt lines were studied, and the ways to improve the manipulating efficiency were investigated. Furthermore, the best manipulating angle range with the highest efficiency was found and verified by simulation.

Since the volume of deposited copper depends on the shape formed by the liquid bridge, when the manipulation angle is selected between 0 degrees and 90 degrees, the cross-sectional area of deposited copper meets the manipulation requirements. The method can meet the stress and strain requirements of the target micro-object. In the electrolysis process, the deposited copper is converted into the electrolyte in a shorter time because the overall volume of the deposition is reduced. When the deposition meets the complete electrolysis from the pipette nozzle to the bottom of the deposition, the expected effect of accurate and nondestructive release is achieved by applying an upward force to the manipulation tool. The electrolyte adhering to the surface of the micro-object evaporates and diffuses into the air at room temperature without causing any pollution to the environment. The release manipulation process is shown in Figure 4.

Therefore, we conducted our study on the basis of the previous research conclusions of research groups. Firstly, based on the finite element simulation model [16], the effect of the manipulating angle of micro-objects represented by microcopper and platinum wires on the manipulating success rate was investigated, and the manipulating strategy to improve the working efficiency was studied. Secondly, the optimal manipulating angle range at the highest efficiency was further identified, and the simulation was verified and corrected in the next step of the study. Finally, the experimental verification of the manipulation strategy was carried out by building a micromanipulation experimental platform. The contributions made here have wide applicability.

## 2. Experimental and Theoretical Research

### 2.1. Experimental Conditions

In this paper, an experimental platform for microstructure manipulation using electrochemical principles is set up, the leading equipment of which includes a desktop vibration isolation platform (Minus K manufacturer, Inglewood, CA, USA), electrostatic meter (Keithley manufacturer, Cleveland, OH, USA), CCD camera and a three degrees of freedom microplatform (THORLABS manufacturer, Newton, NJ, USA), etc. The experimental platform physical diagram is shown in Figure 5.

### 2.2. Experimental Conditions and Method

The pick-and-release manipulation of the microcopper wire with a diameter of 60 μm and a length of 300 μm was verified by using a pipette filled with an electrolyte at a concentration of 0.5 mol/L. The temperature and relative humidity in the laboratory were controlled to be within a stable range before the experiment, and other unnecessary disturbing factors were avoided so as to not cause errors in the experimental results. To achieve the goal of high efficiency and stable picking up, the simulation software was used to model and simulate the micromanipulating system, making the experimental results convincing and reliable.

### 2.3. Theory and Simulation Analysis

Various forces are amplified withinside the microenvironment because of the presence of water molecules within the environment. There is a capillary force among the microcopper wire and the substrate, as proven in Figure 6, which influences the selecting up manner.

There is a certain distance between the microcopper wire and the substrate, which is far less than the radius of the microcopper wire. There is a van der Waals force between the microcopper wire and the substrate, which shows that the two objects attract each other, which also affects the pick-up process. Therefore, we needed to calculate the magnitude of these forces before manipulation. After calculation, there was a big difference between the mass of the microcopper wire and the van der Waals force between the microcopper wire and the substrate. Therefore, the gravity of the microcopper wire was ignored in the subsequent calculation. The force model of the microcopper wire on the substrate is shown in Figure 7.

From the force model, we can understand that once the microcopper wire is at rest, it is subjected to gravity, van der Waals forces [17,18] and capillary forces [19,20,21]. According to Equation (1), the capillary force *F_C_* between the microcopper wire and the silicon substrate can be calculated; Equation (2) can calculate the van der Waals force *F_VDW_* between the microcopper wire and the silicon substrate; Equation (3) can calculate the gravitational force *F_G_* of the microcopper wire. The force relationships to be satisfied during the pick-up manipulation are shown in Equations (4) and (5). The meaning of each variable in Equations (1)–(5) is detailed in [22].
(1)$$F_{C}=\frac{\left(2L+\pi \cdot D)\cdot \sigma \cdot (\cos\alpha +\cos\beta \right)}{1+\frac{d}{H}}$$
(2)$$\begin{array}{l} F_{VDW}=\frac{A_{H}\cdot \sqrt{\frac{D}{2}}}{8\sqrt{2}\cdot d^{\frac{5}{2}}}\cdot L \end{array}$$
(3)$$F_{b}=\sigma_{Cu}\cdot S_{DC}$$
(4)$$F_{b}>F_{VDW}+F_{C}$$
(5)$$\frac{F_{VDW}\cdot L}{\pi (r^{\prime 2}-r_{Nozzle}^{2})}<\delta_{Cu}$$

Because the angle of the pipe is different during micromanipulation, it will affect the shape of the liquid bridge at the deposition, and the shape of the liquid bridge will directly affect the efficiency and quality of the deposition. It is necessary to use mathematical equations such as Young–Laplace [23,24,25] and Kelvin [26,27] to mathematically analyze the effect of the shape of the liquid bridge at different contact angles on the deposition. The boundary conditions of the first-order differential equation system are taken at the end of the micropipetting tube and the upper end of the microsphere, and the boundary conditions can be represented by (6) at the contact between the microsphere and the liquid bridge.
(6)$$\frac{d_{r}}{d_{z}}|{}_{Z_{3}}=\left\{ \begin{array}{ll} {\frac{1}{\tan(\alpha_{3}+\phi_{1})}}^{} & (\alpha_{3}+\phi_{1})\neq \frac{\pi }{2}\\ 0 & (\alpha_{3}+\phi_{1})=\frac{\pi }{2} \end{array}\right.$$

The boundary conditions at the end of the liquid bridge and the microtube are shown in Formula (7):(7)$$\frac{d_{r}}{d_{z}}|{}_{Z_{1}}=\left\{ \begin{array}{ll} -1/\tan\alpha_{1} & \alpha_{1}\neq \frac{\pi }{2}\\ 0 & \alpha_{1}=\frac{\pi }{2} \end{array}\right.$$
where *α*_1_ and *α*_2_ are the upper and lower contact angles between the liquid bridge and the pipette and the liquid bridge and the microcopper wire. Figure 8 is a side view of the liquid bridge deposition model. Figure 9 shows the front view of the liquid bridge deposition model (in the case of an oblique angle of 45°). *α*_1_ is the filled angle of the microcopper wire in the liquid bridge.

This experiment was based on the study of the factors affecting the efficiency of microscale electrochemical deposition and electrolysis, as well as the effects of the solution concentration, ambient relative humidity and other factors on the rate and quality of microscale electrochemical deposition and electrolysis. The establishment of microscale models and simulation methods was investigated. Based on the theoretical model [28], an FEM of the curved liquid surface was established. The simulation analysis of pick-up, transfer and release under different manipulation angles was performed to verify the correctness of the theoretical model establishment. Therefore, a comprehensive comparative analysis of the existing simulation software was carried out starting from the microscale modeling and simulation methods. On the basis of the finite element simulation results, the above model and related parameters were verified by several practical experiments until reliable pick-up, stable transfer and accurate nondestructive release were achieved.

## 3. Results and Discussion

### 3.1. Dynamic Simulation

In the simulation, a pipette filled with an electrolyte was used to manipulate the microcopper wire with a diameter of 60 μm and a length of 300 μm. Based on the study mentioned above the of pipette and microcopper wire sandwich angle of 90°, we selected the pipette manipulation point at the center point of the microcopper wire, the stress condition of deposited copper and the model change of sedimentation when the pipette and microcopper wire angle was 30°, 45° and 60°. Because the evolution of the contact angle would cause the corresponding modifications in capillary force, liquid bridge volume, and shape, this paper combined its calculations with Young–Laplace square according to Kelvin thermodynamics. By modeling the liquid bridge at different contact angles by combining the calculation of the elliptic integral equation, Figure 10 shows the manipulation contact diagram of different picking angles.

The size of the manipulated object is at the micron level, and various forces are amplified under the microenvironment [29,30]. The material of the pipette tube is quartz glass; a quenching treatment during the production process, so that the most vulnerable nozzle antibending and impact strength can be improved [22,31], in the course of manipulation can ensure that the pipette has sufficient strength not to be damaged. However, because the process of the deposition of copper will be affected by many factors [32,33], such as environmental relative humidity, sediment concentration and current density and other external factors, as well as pick-up angle and pick-up position and other physical factors, the sediment quality has other differences [34,35]. This directly or indirectly affects the effectiveness of the pick manipulation, so it is necessary to analyze the stresses on the deposited copper site before simulating the actual manipulation. In this paper, the stress of deposited metal copper at the pipette nozzle was simulated using finite element software. The pipetting tube was full of an electrolyte, and the nozzle diameter was 15 μm; the microcopper wire diameter was 60 μm, and the length was 300 μm. Its model and the simulation results are shown in Figure 11.

Figure 11 shows that the stress range in the peripheral part of the deposited copper is relatively large and shows the continuous diffusion of water molecules in the solution to the outside environment during the sedimentation process. The decreasing state of water molecule concentration makes the quality of deposited copper relatively poor compared with the sedimentary interior. Based on this theory and simulation results, it is very important to study the optimal manipulating angle of successful pick-up, transfer and release to realize the efficient three-dimensional manipulation of micrometallic components.

Because the model is more complex, the model needs to be divided into mesh, and we need to define the material properties, according to the displacement boundary conditions in elasticity and the actual characteristics of the model. This model is fixed with six spatial degrees of freedom. To limit the relative rigid body displacement, the boundary condition is set to five nonvertical constraint degrees of freedom to be completely fixed, and the picking simulation diagram is as shown in Figure 12.

From Figure 12, it can be concluded that the electrolyte in the pipetting tube chemically produced a connection between the deposited metal copper, pipetting pipet and the microcopper wire. At this time, the pipetting tube and micrometal copper wire became one whole, through the pipetting tube at one end of the application of load force to overcome the microcopper wire and the base between the van der Waals force and capillary force and its gravity, to achieve a fast and stable pick-up effect. Using simulation software to stress the analysis and simulation of the deposited metal copper and the local amplified stress diagram obtained by defining the load and boundary constraints, we created the simulations shown in Figure 13. After the electrolyte deposition, the stress value in the connection bending of the pipette nozzle is at its maximum during the micromanipulation picking-up process.

Sampling and simulation experiments are carried out at the maximum stress of deposited copper in Figure 13. The simulated load force is applied in the vertical direction, and the load force calculation formula is used as the load force to the lower end of the microcopper wire in the model; the stress change curve of the metal copper deposited during manipulation was obtained as shown in Figure 14.

The capillary force is greatly affected by the volume and contact angle of the liquid bridge. From the simulation result curve shown in Figure 14, when picking up manipulation, when the contact angle is small (30°), the stress of the copper deposit is about 250 Mpa, which exceeds the tensile strength range of the copper deposit in the experiment, about 240 mpa, which leads to damage of the copper deposit, which affects the experimental effect. When the contact angle is large (60°), according to the simulation results, it can be seen that the maximum stress on the deposition site is about 120 Mpa. However, the capillary force increases with the volume of the liquid bridge and then decreases gradually as the volume increases, which leads to the increased complexity of the analysis during manipulation due to the consideration of changes in the capillary force. In contrast, when the volume of the liquid bridge is specific, the capillary pressure of the liquid bridge decreases with the increase of the contact angle. When the pipetting tube is perpendicular to the microcopper wire (90°), it is assumed that, in the pipette–liquid bridge, the liquid bridge substrate contact angle is equal (*α*_1_ = *α*_3_, *α*_2_ = *α*_4_), so the capillary force in the process of picking can be negligible. Although the stress of the deposited copper is relatively minimal, considering the release manipulation, because of orientation problems with the actual manipulation of the Microvision equipment, the liquid bridge volume, the deposition copper electrolysis time and other factors, we had to ensure that the pipette contact angle was not too large when manipulating in order to make the release manipulation successful and efficient.

Therefore, combined with the simulation results of this paper, with the actual pick-up and release manipulation, the pipette and microcopper wire contact angle is about 45° for the best manipulating contact angle range. For the selection of the manipulation point, in this paper, the simulation research was carried out at different contact angles of the microcomponent manipulation center and focused on the influence of the manipulation angle on the actual manipulation experiment. However, the selection of the manipulation point is closely related to the size of the manipulation object, the shape of the liquid bridge that produces the capillary force and other factors. Therefore, in the following study, we focus on analyzing the impact of different manipulation points of the pick-up and release manipulation and finding the relationship between the selection of the manipulation position and the pick-up and release manipulation in other cases.

### 3.2. Experimental Verification

#### 3.2.1. Micromanipulation Experiment Verification at Different Manipulating Points

In the experiments, a microcopper wire of the same radius and 100 μm in length was used as the target. The midpoint of the microcopper wire, four-fifths of the total length of the microcopper wire and the edge of the microcopper wire were taken as the boundary points for the pick-up and release manipulation. By comparing the experimental results of several control variable groups and using the overall manipulation completion time as an indicator, the manipulation point with the best manipulative effect and the interval of the best manipulating point was identified. The experimental results for each manipulation point manipulation process are shown in Figure 15.

According to the experimental results in Figure 15, the following conclusions can be obtained: when the pipette filled with the electrolyte is located at the midpoint of the microcopper wire, the target can be firmly picked up and transferred to the designated position by deposition and then released. The average time of sediment during the whole process of picking up is about 190 s, the average time of electrolysis is about 360 s and the overall manipulating time is controlled within 10 min. When the pipette was positioned at four-fifths of the total length of the microcopper wire, Figure 15b shows that as the pipette moved upward during the pick-up process, the pipette could also achieve a good pick-up effect. Due to the adhesion effect, the adhesion force at the end picked up away from the deposition is significantly greater than the part near the deposited copper, resulting in a bending trend on one side of the microcopper wire during the pick-up process. There was apparent damage to the deposited copper.

However, the tendency of the microcopper wire to drop on one side during the release process causes the released side to contact the silicon substrate first, which closes the overall circuit, and the electrolysis process begins. With the other side of the slow fall in progress, the electrolysis process synchronized. When the other side of the microcopper wire contacts the silicon substrate, waiting for the complete electrolysis can also be completed after the completion of the nondestructive pick-up and release micromanipulation experiment. In the conclusion of the experiment, it can be shown that the time used for deposition is approximately the same as when the pipette is located at four-fifths of the microcopper wire and the midpoint, but the time used after the release process is about 10 s less than that between the centers of the previous group. When the manipulation point of the pipette is located at the edge of the microcopper wire, it can be seen from Figure 15c that the side closest to the deposition can be picked up slightly for some distance due to the deposit at the pipette nozzle. However, with the increase of upward displacement of the pipette, the microcopper wire will be dropped as shown in Figure 15c, and the deposited copper will be damaged due to the uneven severe force, and thus the pick-up manipulation will fail.

#### 3.2.2. Micromanipulation Experiment Verification at Different Manipulating Angle

To verify the efficiency of the electrochemical micromanipulation method and the reliability of simulation results, the pipetting tube filled with an electrolyte was used to carry out the actual pick-up and release experimental verification of microcopper wire with a diameter of 15 μm and a length of 300 μm. The pipette tilt angle of the electrolyte-filled tube was set to between 45° and 60°. The average time to pick-up and release microcopper wires with a length of 300 μm was 182 s and 298 s, respectively, by the electrochemical principle. The microcopper wire was stable and lossless during the pick manipulation, and the electrolysis was entirely and pollution-free during the release manipulation. The pick-up and release experimental manipulation are shown in Figure 16.

### 3.3. Experimental Result

Table 1 shows the pick-up and release times for the five experimental groups. The data in the table show that the overall manipulation time is about 10 s longer when the pipette is in the middle position than when it is off the middle point. The releases all failed when the pipette was located at the edge of the copper wire. However, considering the purpose of not changing the original form of the object and not damaging the object, the middle point was chosen as the best manipulating point.

Since the angle between the nozzle of the pipette and the microcopper wire was 90° when searching for the best manipulating point, the experiments of the contact angle-based manipulation strategy only required the tilt angle of the electrolyte-filled pipette to be set between 45° and 60°. Several sets of pick-and-release experiments were performed on the microcopper wire with a length of 300 μm by electrochemical methods, as shown in Table 2. The average pick-up and release times were 182 s and 298 s, respectively, and the copper wire was stable and undamaged during the pick-up manipulation, and the deposited copper was completely electrolyzed and free of contamination during the release manipulation. The pick-up and release manipulation times based on different micromanipulation angles are shown in Table 2.

The experimental results show that the contact angle between the pipette and microcopper wire is 45° to 60°. The pick and release manipulation can be realized successfully, and the manipulative effect is the best. By comparing the experimental results and analysis, it is clear that the two critical strategic indicators that affect the manipulation efficiency and effect are the selection of the manipulating point and the manipulation angle. They affect the picking and releasing accuracy of the micromanipulation. For the pick-up manipulation, the performance of picking up the microcopper wire differs when the pipette is located at different manipulation points, and the shortest pick-up time is shown in the case where the pipette is located at four-fifths of the length of the microcopper wire. In addition, combined with a comprehensive analysis of the manipulating angles, the different manipulating angle ranges produced different deposition shapes and smaller contact angles during pick-up did not pick up the target object well. For the release manipulation, when the manipulation point is chosen to be off-center, the manipulation time can be shortened by releasing a section that touches the silicon substrate first and slowly drop the other end. However, combining the selection of the manipulating angle with the objective factors such as particular manipulation tasks, the center point is taken as the best manipulating point, and the best manipulation angle range between 45° and 60° is used for micromanipulation.

## 4. Conclusions

This paper provides an implementable idea for different types of metal micromanipulation and promotes the rapid application of micromanipulation techniques in microelectromechanical systems. This study based on electrochemically assisted methods to achieve manipulation of microcopper wires led to the following main conclusions:The average time to complete the manipulation was 182 s and 298 s when the pipette angle was between 45° and 60°. The most efficient pick-and-place manipulation was performed for copper wires with a length of 300 μm.When the manipulation point was selected at the four-fifths position of the copper wire, the total manipulation time was about 540 s; when the manipulation point was selected at the middle position of the copper wire, the total manipulation time was about 551 s. When the manipulation point is in the middle position, there is almost no change in the trait structure change of the metal. Considering the actual production needs, the middle point was selected as the most suitable position for the manipulation strategy.The object manipulated in this paper was copper wire, and the strategy proposed in this paper can be applied to the micron-level metal manipulation of different materials. When the hardness of other metals meets the actual manipulation requirements, the manipulating position selected can deviate from the center point. The overall manipulation time is shortened by about 10~15 s compared to the center point position.

## 5. Patents

The authors declare that they have no known competing financial interests or personal relationships that could have appeared to influence the work reported in this paper.

## Figures and Tables

**Figure 1 micromachines-13-02151-f001:**
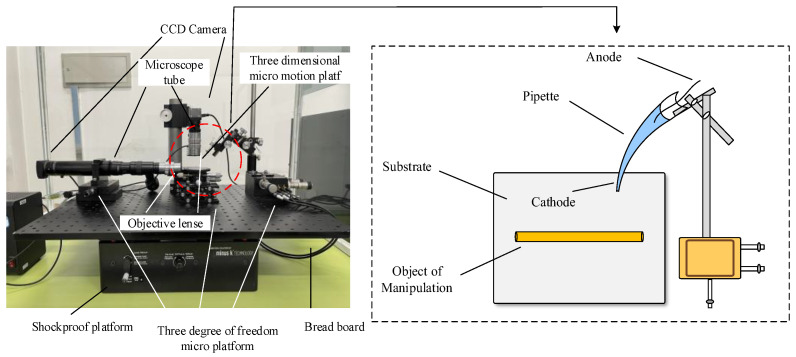
Electrochemical-based metal microstructure manipulating platform diagram.

**Figure 2 micromachines-13-02151-f002:**
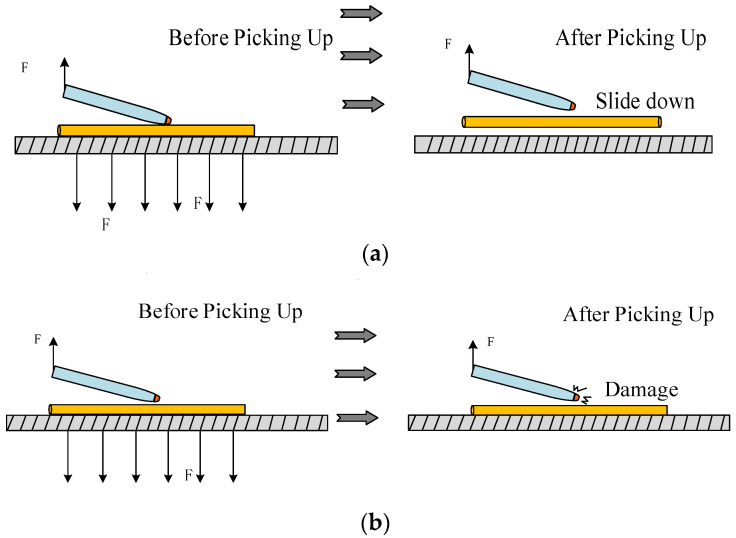
Slips or damage caused by the improper selection of manipulating angles. (**a**) Microcopper wire slip situation; (**b**) damage of deposited copper.

**Figure 3 micromachines-13-02151-f003:**
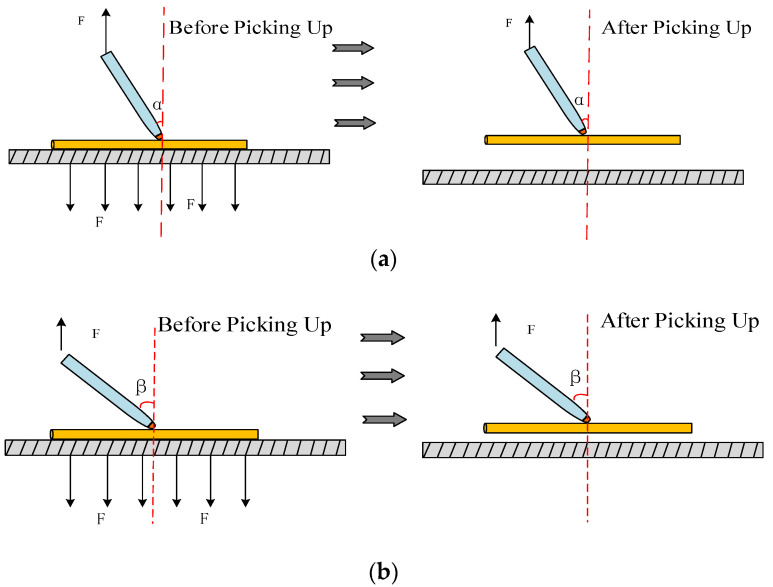
Schematic diagram of successful manipulation of different manipulation points. (**a**) Pick when the pipette tilt angle is α; (**b**) pick when the pipette tilt angle is β.

**Figure 4 micromachines-13-02151-f004:**
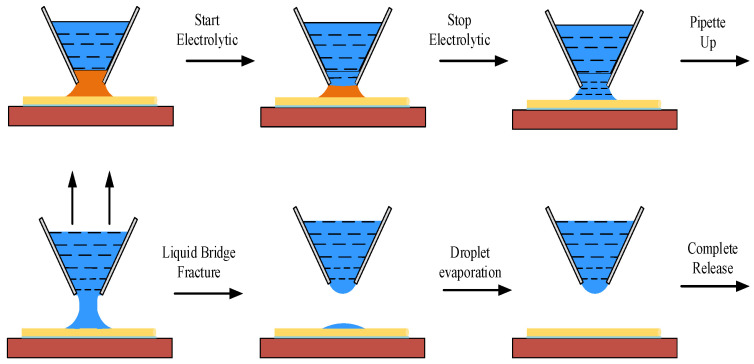
Manipulation flow chart of electrochemical electrolytic release.

**Figure 5 micromachines-13-02151-f005:**
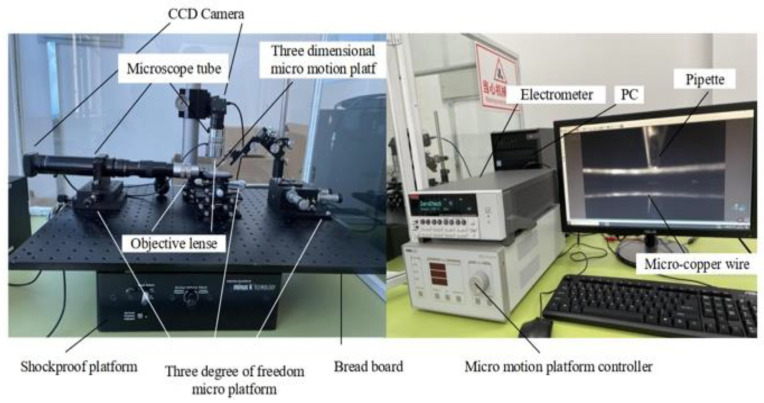
Physical drawing of the experimental platform.

**Figure 6 micromachines-13-02151-f006:**
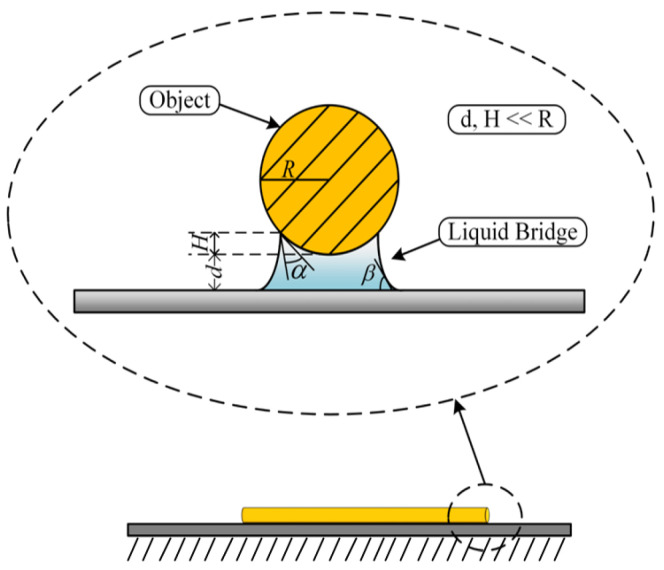
Schematic diagram of microliquid bridge between substrate and object.

**Figure 7 micromachines-13-02151-f007:**
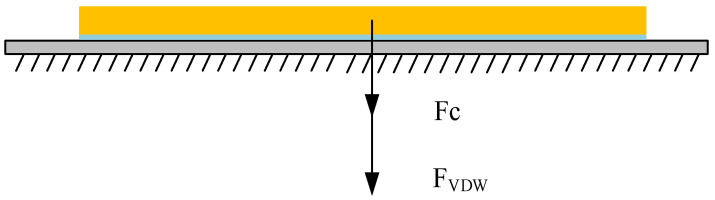
Force analysis of microcopper wire.

**Figure 8 micromachines-13-02151-f008:**
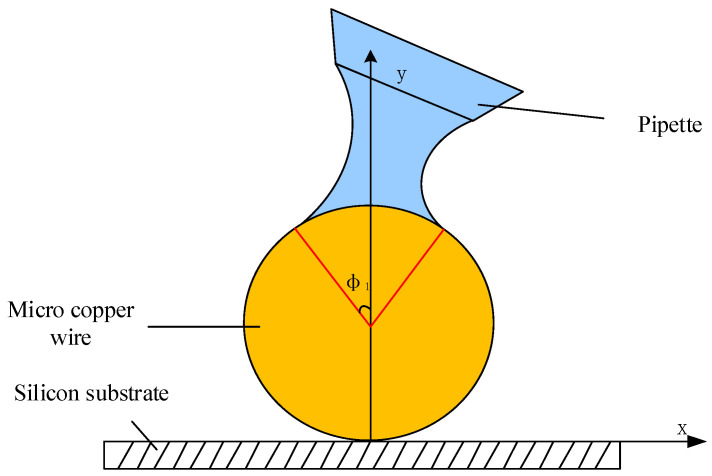
Side view of the liquid bridge model.

**Figure 9 micromachines-13-02151-f009:**
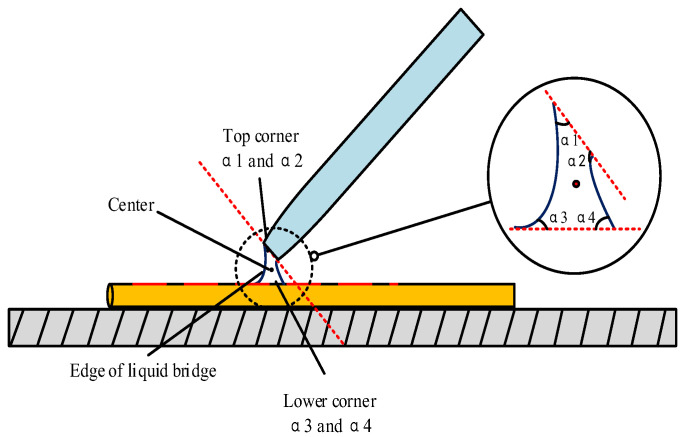
Liquid bridge model diagram.

**Figure 10 micromachines-13-02151-f010:**
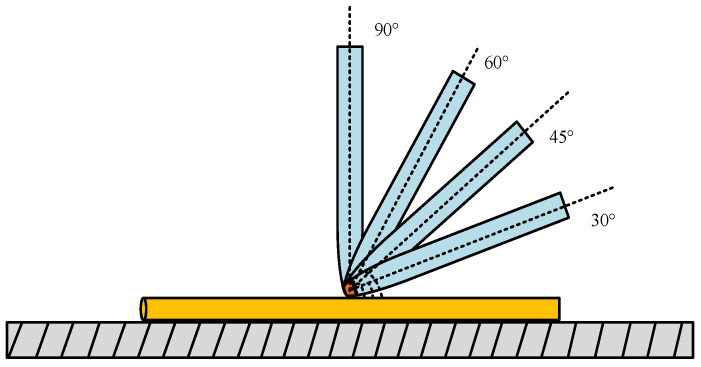
Pick contact diagrams from different angles.

**Figure 11 micromachines-13-02151-f011:**
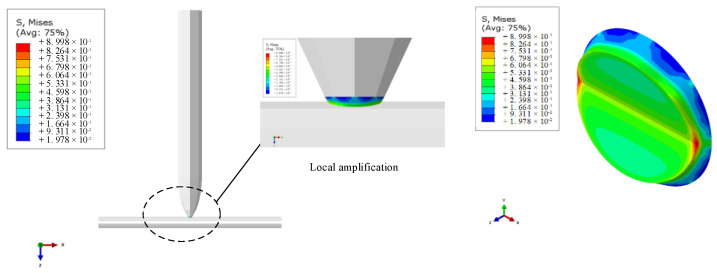
Local stress nephogram at deposited copper.

**Figure 12 micromachines-13-02151-f012:**
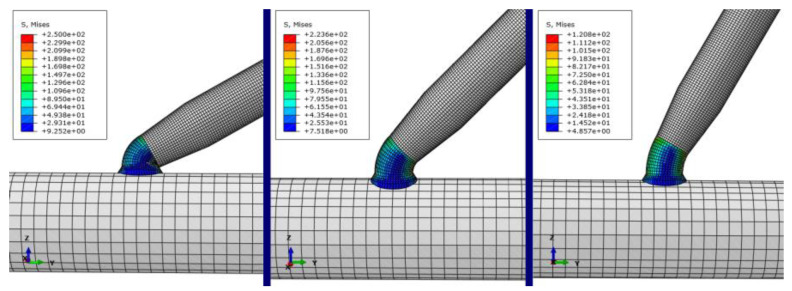
Pick–release manipulation model diagrams at different contact angles.

**Figure 13 micromachines-13-02151-f013:**
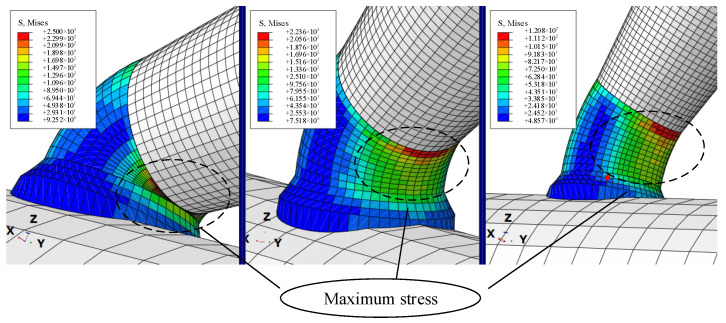
Simulation of the stress on copper deposited at different angles.

**Figure 14 micromachines-13-02151-f014:**
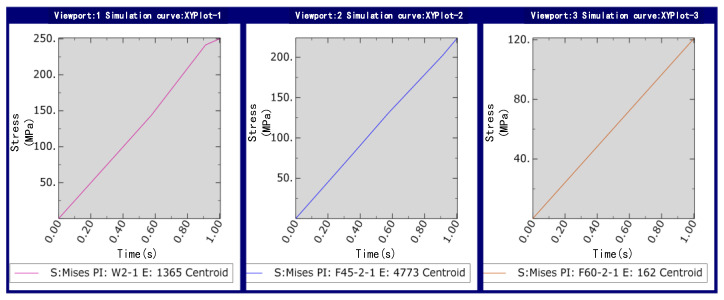
Stress change map at a different tilt angles.

**Figure 15 micromachines-13-02151-f015:**
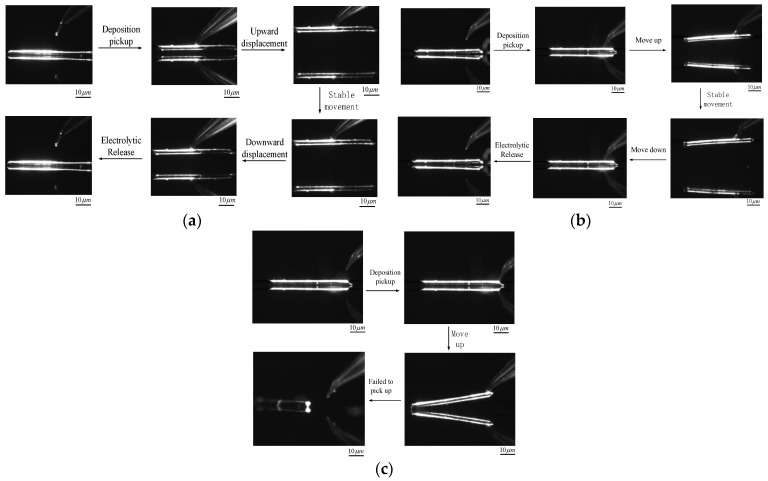
Micromanipulation experiments at different manipulating positions. (**a**) The manipulation point is the middle point; (**b**) the manipulating point is at 4/5 of the total length; (**c**) the manipulating point is at the edge.

**Figure 16 micromachines-13-02151-f016:**
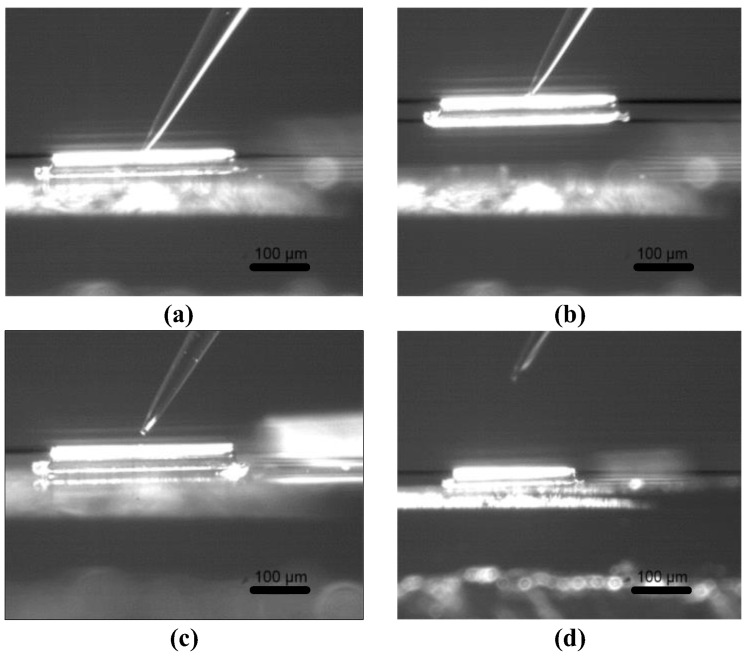
Experimental results of successful manipulation. (**a**) Deposition; (**b**) pick-up; (**c**) electrolysis; (**d**) release.

**Table 1 micromachines-13-02151-t001:** Time consumption table of micromanipulation at different manipulating position.

Manipulation Time (s)		Copper Wire Midpoint	Copper Wire at 4/5	Copper Wire at the Edge
Group 1	Pick-up time	192	196	215
	Release time	358	345	/
	Manipulation time	550	541	/
Group 2	Pick-up time	185	179	175
	Release time	366	345	/
	Manipulation time	551	524	/
Group 3	Pick-up time	203	215	211
	Release time	378	356	/
	Manipulation time	581	571	/
Group 4	Pick-up time	175	183	194
	Release time	353	345	/
	Manipulation time	528	528	/
Group 5	Pick-up time	186	193	206
	release time	362	345	/
	Manipulation time	548	538	/

**Table 2 micromachines-13-02151-t002:** Time consumption of micromanipulation at different manipulating angles.

Manipulation Time (s)		t < 45°	45° < t < 60°	t = 90°
Pick-up time	Group 1	175	181	223
	Group 2	178	179	236
	Group 3	169	186	219
	Average	174	182	226
Release time	Group 1	/	311	378
	Group 2	/	286	364
	Group 3	/	297	372
	Average	/	298	371

## Data Availability

Not applicable.

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
