# Peer review of "Study on the Manipulation Strategy of Metallic Microstructures Based on Electrochemical-Assisted Method"

_micromachines, 2022, doi:10.3390/mi13122151_

Round 1
Reviewer 1 Report
Li et al. have used an electrochemical-assisted method to study the manipulation strategy of metallic microstructures. There are a few typographic and grammatical errors, such as issues with spacing, commas, incorrect upper- and lower-case usage, spelling mistakes, and tenses. To improve the overall quality and presentation of the manuscript, vigorous revision (major revision) is required. Consider the following comments in this regard, which may be useful in improving the manuscript.
1. The introduction covers a lot of literature. The authors must state the research gap. Furthermore, they should be briefly discussed in terms of their benefits and drawbacks.
2. Given that the introduction contains descriptions of miniature manipulation and rheology it is useful to see competitive methods used in micromanipulation such as microfluidics and 3D printing for a variety of applications. Please consider the following references and include them in the manuscript. Example, Doi: 10.1016/j.mtadv.2022.100205; 10.1016/j.ijmecsci.2022.107805; 10.1021/acs.macromol.2c00052; 10.1002/admt.202101347; 10.3390/bios12040220.
3. Authors must provide a brief statement describing the novelty, purpose, and scope of work.
4. Why did the authors consider using micro-Cu and Pt lines in the study?
5. Line no. 281, Vandervoorne force should be ‘Van Der Waals force’.
6. In line 284, the terms shown and shows are used in the same sentence. Similar errors can be found throughout the manuscript.
7. Is it possible to investigate the viscoelastic behavior at various manipulating positions, as well as the storage modulus and loss modulus, in relation to oscillation strain rheological studies?
8. Please rewrite the conclusion section.
Author Response
Response to Reviewer 1 Comments:
We appreciate the time and effort you’ve spent in reviewing our manuscript " Study on the Manipulation Strategy of Metallic Microstructures Based on Electrochemical Assisted Method ". Thanks very much for the reviewer’s comments and suggestions. These comments are very valuable for improving our manuscript. We have checked and considered these comments carefully and have revised our manuscript correspondingly. All the changes of the manuscript and the responses to the reviewers are listed below. In addition, the changes are highlighted in the updated manuscript.
We are willing to make further changes if you feel them necessary and we are grateful for your advices on the matter.
Thanks very much!
Yours Sincerely
Dongjie Li
Reviewer 1: Comments and Suggestions for Authors:
Li et al. have used an electrochemical-assisted method to study the manipulation strategy of metallic microstructures. There are a few typographic and grammatical errors, such as issues with spacing, commas, incorrect upper- and lower-case usage, spelling mistakes, and tenses. To improve the overall quality and presentation of the manuscript, vigorous revision (major revision) is required. Consider the following comments in this regard, which may be useful in improving the manuscript.
Point 1: The introduction covers a lot of literature. The authors must state the research gap. Furthermore, they should be briefly discussed in terms of their benefits and drawbacks.
Response 1: Thanks very much for the reviewer’s comment. The current status of research on various modalities of micromanipulation and the research gaps have been added in the manuscript at the appropriate locations [Page 2, line 54-59]. The advantages and disadvantages between them are also briefly discussed [Page 2, line 64-65, 70-72, 75-76, 79-80, 83-87].
Point 2: Given that the introduction contains descriptions of miniature manipulation and rheology it is useful to see competitive methods used in micromanipulation such as microfluidics and 3D printing for a variety of applications. Please consider the following references and include them in the manuscript. Example, Doi: 10.1016/j.mtadv.2022.100205; 10.1016/j.ijmecsci.2022.107805; 10.1021/acs.macromol.2c00052; 10.1002/admt.202101347; 10.3390/bios12040220.
Response 2: Thanks very much for the reviewer’s advice. We have added the relevant literature [1-3, 9-10] and described it briefly in the appropriate places in the manuscript. The modified sections are highlighted in the updated manuscript. [Page 2, line 54-59]
References:
- Bendre, A.; Bhat, P.; Lee K.H.; Altalhi, T.; Alruqi, M.A.; Kurkuri, M. Recent Developments in microfluidic technology for synthesis and toxicity-efficiency studies of biomedical nanomaterials. Mater Today Adv, 2022, 13,1-17.
- Bhat, M.; Thendral, V.; Uthappa, U.T.; Lee, K.H.; Kigga, M.; Altalhi T.; Kurkuri M.D.; Kant, K. Recent Advances in microfluidic platform for physical and immunological detection and capture of circulating tumor cells. Biosensors, 2022, 12,27-37.
- Mahmud, M. A. P.; Bazaz, S.R.; Dabiri, S.; Mehrizi, A.A.; Asadnia, M.; Warkiani, M.E.; Wang, Z.L. Advances in MEMS and microfluidics-based energy harvesting technologies. Adv. Mater. Technol, 2022, 7,7-14.
- Liu, G.S.; Bhat, M.P.; Kim, C.S.; Kim, J.; Lee, K.H. Improved 3D-printability of cellulose acetate to mimic water absorption in plant roots through nanoporous networks. Macromolecules, 2022, 55, 1855-1865.
- Zheng, Z.P.; Zhang, J.F.; Feng, P.F.; Wang J.J. Controllable fabrication of microstructures on the metallic surface using oblique rotary ultrasonic milling. Int. J. Mech. SCI, 2022.
Point 3: Authors must provide a brief statement describing the novelty, purpose, and scope of work.
Response 3: Thanks very much for the reviewer’s comment. We have provided a short statement in the manuscript to describe the novelty, purpose and scope of the work. and has been highlighted in the updated manuscript [Page 2, line 89-95].
Point 4: Why did the authors consider using micro-Cu and Pt lines in the study?
Response 4: Thanks very much for the reviewer’s comment. The method proposed in this manuscript is currently applicable to micro-operational objects of conductive metals, which can be used for gold, silver, aluminum, nickel, cobalt, tin, platinum, etc. Theoretical studies cover all metals, while practical experiments focus on the most commonly used conductive metals Cu and Pt. In addition, we have conducted preliminary studies related to additive manufacturing by electrochemical deposition on non-conductive materials, which allows the deposition of growth between different materials by dynamic electrochemical deposition methods. Therefore, the method in the manuscript can be successfully applied to non-conductive materials.
Point 5: Line no. 281, Vandervoorne force should be ‘Van Der Waals force’.
Response 5: Thanks very much for the reviewer’s comment. The manuscript has been carefully checked and corrected for writing errors in the manuscript.
Point 6: In line 284, the terms shown and shows are used in the same sentence. Similar errors can be found throughout the manuscript.
Response 6: Thanks very much for the reviewer’s comment. We have carefully checked and revised the manuscript for English grammar, and writing errors, and have highlighted the corrections in the manuscript.
Point 7: Is it possible to investigate the viscoelastic behavior at various manipulating positions, as well as the storage modulus and loss modulus, in relation to oscillation strain rheological studies?
Response 7: This manuscript focuses on the analysis of the forces during the pick-up and release of the micro-metallic wire as a typical research object. A theoretical force model of the main forces on the micro-metallic wire and the pipette during the pick-up and release of the micro-constituents using electrochemical methods under ideal conditions is established, and a preliminary analysis of the forces during manipulation is presented. The stress-strain applied at the deposited copper is mainly used as a measure to derive the optimal manipulation position. As for the viscoelastic behavior, the storage modulus and the loss modulus are not considered in this manuscript study for the time being.
Point 8: Please rewrite the conclusion section.
Response 8: Thanks very much for the reviewer’s advice. We have revised the conclusion section of the manuscript to refine some of the key findings. and have highlighted them in the updated manuscript. [Page 13, line 432]

Reviewer 2 Report
Overall comments:
The English language is very bad. It was very difficult for me to follow the text.
line 19 - use "the finite.." instead of "a finite.."
line 23 - a typo mistake ..."transferring"
line 23 - "are" instead of "is"
line 28 - add a space before "This..."
line 39 - I didn't catch the
meaning of the last words "and different total its gathering and bundling."
line 45 - put "have" instead of "has"
line 47 - put "have" instead of "has"
line 59 - maybe you should add "which can.."
line 74 - please redo the whole sentence...I can't catch the meaning of it.
line 77 - Instead of "Because of the above" maybe use "Considering the above results "
line 78 - typo mistake...put "dot" instead of "comma" after [11].
line 89 - Add explanations to the second image (right side) of figure 1.
line 101 - please redo de whole sentence.
line 104 - when did you explain the simulation model ...please change the sentence "...based on the above simulation model,.."
line 115 - please define "moderately" add some values. the meaning is very confused for me.
line 129 - please redo this phrase - "Therefore, for the electrochemical-based metal microstructure control method previously proposed by our group.".
line 130 - "the finite element simulation model" please add a reference for "the finite element simulation model" or a short description of it.
line 147 - The description of the "Experimental method" is very, very short and the parameters (e.g. temperature, humidity) are not correlated with the expected results.
line 152 - what means "to affect the experimental results. "???
line 157 - remove the dot between " micro-environment. Because"
line 176 - change "pressure" with "force" for consistency.
lines 178 to 182: the whole paragraph "Equation (3) can calculate the gravitational force FG of the micro copper wire. The force relationships to be satisfied during the pick-up manipulation are shown in Eq. (4) and Eq. (5). The detailed meaning of each variable in Eqs. (1-5) has been explained in earlier research papers of the subject group, as detailed in the literature [26]." should be reformulated. I didn't understand the meaning of this words " earlier research papers of the subject group".
line 201 - put "dot" before "Figure 8..."
lines 202 - 204 - switch the order of these two sentences and use the same symbol in the explanation likes in figure 9.
lines 209 - 228 - These phrases are too long and its difficult for me to follow the meaning "Based on the study of the factors affecting the efficiency of microscale electrochemical
deposition and electrolysis, as well as the effect of critical factors such as voltage, solution
concentration, and relative humidity of the environment on the velocity and quality of
microscale electrochemical deposition and electrolysis, the method of microscale model
establishment and simulation is studied. ....".
lines 231 - 236 - these are not results...are a description and the chapter title is: "dinamic simulation results".
line 261 - figure 6 doesn't show the stress range.
line 282- the sentence "Using simulation software to stress
analysis and simulation of deposited metal copper, the resulting partial amplification of
stress as shown in Figure 13 shows." is a bit ambiguous...for example "...the resulting partial amplification of
stress as shown in Figure 13 shows."
line 313 - the sentence is a bit weird, may be you can change it.
line 326 - put "pick-up" instead of "pick".
line 329 - maybe use "length" instead of "size"
line 331 - "were" instead of "was"
page 11 - for consistency please use "manipulation point" instead of "manipulating point".
line 346 - which figure?
line 351 - what means 10 mins?
line 374 - subchapter 3.2.1 is the same as subchapter 3.2.2 ...the titles are idem and the contents are almost the same.
line 375 - typo mistake "the" instead of "thr"
line 388 - please explain the characteristics of each experimental group. In table 1 there are 5 groups but I didn't see the differences between them.
line 403 - small letter at "Electrolyzed"
line 409 - It is a non-sense in " By comparing the experimental results and
analysis," should be reformulated.
line 412 - instead of "It is affect" seems to be "It affects"..
lines 407-430 - appear to be conclusions. The paragraph doesn't say important things, the ideas are also found above. From my point of view, this paragraph should be removed or totally change.
line 434 - it seems that you are misssing a verb "is based" instead of "based".
line 452 - the phase is too long. Please redo it, split it into 2 or 3 sentences for more consistency.
Author Response
Response to Reviewer 2 Comments:
We appreciate the time and effort you’ve spent in reviewing our manuscript " Study on the Manipulation Strategy of Metallic Microstructures Based on Electrochemical Assisted Method ". Thanks very much for the reviewer’s comments and suggestions. These comments are very valuable for improving our manuscript. We have checked and considered these comments carefully and have revised our manuscript correspondingly. All the changes of the manuscript and the responses to the reviewers are listed below. In addition, the changes are highlighted in the updated manuscript.
We are willing to make further changes if you feel them necessary and we are grateful for your advices on the matter.
Thanks very much!
Yours Sincerely
Dongjie Li
Reviewer 1: Overall comments:
Point 1: The English language is very bad. It was very difficult for me to follow the text.
Response 1: Thanks very much for the reviewer’s comment. The manuscript has been carefully checked and corrected for English grammar, difficult-to-understand parts and writing errors in the manuscript have been carefully checked and corrected, and the corrections have been highlighted in the manuscript.
Point 2: line 89 - Add explanations to the second image (right side) of figure 1.
Response 2: Thanks very much for the reviewer’s comment. It has been interpreted for the right side of figure 1, and added in the updated manuscript [Page 2, line 86-87].
Point 3: line 130 - "the finite element simulation model" please add a reference for "the finite element simulation model" or a short description of it.
Response 3: Thanks very much for the reviewer’s advice. We have added the relevant literature and described it briefly in the appropriate places in the manuscript. The modified sections are highlighted in the updated manuscript [Page 4, line 128-129].
Point 4: line 147 - The description of the "Experimental method" is very, very short and the parameters (e.g. temperature, humidity) are not correlated with the expected results.
Response 4: Thanks very much for the reviewer’s comment. For the experimental methods section: we have added the necessary descriptions in the corresponding positions [Page 5, line 148-150]. For the parameters (e.g., room temperature, ambient relative humidity): Since this experiment focuses on the strategic issue of micro-metal manipulation, the experiment requires controlling the extraneous variables in the appropriate range is sufficient, but needs to be taken into account among the factors affecting the experimental results. So it needs to be mentioned in the article, and the corrections have been highlighted in the manuscript.
Point 5: lines 202 - 204 - switch the order of these two sentences and use the same symbol in the explanation likes in figure 9.
Response 5: Thanks very much for the reviewer’s advice. We have switch the order of corresponding sentences and use the same symbol in the explanation in figure 9. The modified sections are highlighted in the updated manuscript [Page 6, line 199-205].
Point 6: line 388 - please explain the characteristics of each experimental group. In table 1 there are 5 groups but I didn't see the differences between them.
Response 6: Thanks very much for the reviewer’s advice. We have added an explanation of the experimental data in table 1 at the corresponding location, and highlighted it in the updated manuscript [Page 12, line376-379].
Point 7: lines 407-430 - appear to be conclusions. The paragraph doesn't say important things, the ideas are also found above. From my point of view, this paragraph should be removed or totally change.
Response 7: Thanks very much for the reviewer’s advice. The necessary deletions have been made to lines 407-430, but because of the need for the necessary explanatory notes on the experiments, including a description of the specific choices of manipulation strategies, they are discussed separately in terms of both pickup and release operations. However, in the conclusion section is a quantitative summary of the experimental results obtained above as well as the phenomena and the necessary outlook for future research.

Round 2
Reviewer 1 Report
The revision is satisfactory, however, I recommend for the minor spell and grammar check though out the manuscript.
Reviewer 2 Report
The last version of the "Study on the Manipulation Strategy of Metallic Microstructures Based on Electrochemical Assisted Method" manuscript took care of all my suggestions/comments.
One more comment: line 437 - "data" instead of "data".
From my point of view, the manuscript should be published.